# Towards Flexible 3D Perception: Object-Centric Occupancy Completion Augments 3D Object Detection

**Chaoda Zheng**[1,2*]    **Feng Wang**[3]    **Naiyan Wang**[4]    **Shuguang Cui**[2,1]    **Zhen Li**[2,1]

[1]FNii-Shenzhen    [2]SSE, CUHK-Shenzhen    [3]TuSimple    [4]Xiaomi EV

`{feng.wff, winsty}@gmail.com`

`{chaodazheng@link., shuguangcui@,lizhen@}cuhk.edu.cn`

https://github.com/Ghostish/ObjectCentricOccCompletion

## Abstract

While 3D object bounding box (bbox) representation has been widely used in autonomous driving perception, it lacks the ability to capture the precise details of an object's intrinsic geometry. Recently, occupancy has emerged as a promising alternative for 3D scene perception. However, constructing a high-resolution occupancy map remains infeasible for large scenes due to computational constraints. Recognizing that foreground objects only occupy a small portion of the scene, we introduce object-centric occupancy as a supplement to object bboxes. This representation not only provides intricate details for detected objects but also enables higher voxel resolution in practical applications. We advance the development of object-centric occupancy perception from both data and algorithm perspectives. On the data side, we construct the first object-centric occupancy dataset from scratch using an automated pipeline. From the algorithmic standpoint, we introduce a novel object-centric occupancy completion network equipped with an implicit shape decoder that manages dynamic-size occupancy generation. This network accurately predicts the complete object-centric occupancy volume for inaccurate object proposals by leveraging temporal information from long sequences. Our method demonstrates robust performance in completing object shapes under noisy detection and tracking conditions. Additionally, we show that our occupancy features significantly enhance the detection results of state-of-the-art 3D object detectors, especially for incomplete or distant objects in the Waymo Open Dataset.

## 1 Introduction

In autonomous driving, accurate and robust 3D scene perception is crucial for safe and efficient navigation. Conventional perception systems primarily adopt 3D object bounding boxes as the perception representation [25, 6, 15, 16]. However, the limitations of 3D bounding boxes (bboxes) are becoming increasingly pronounced as the demands for perception accuracy continue to escalate. Since a 3D bbox is essentially a cuboid that encapsulates the object, it fails to capture the precise details of the object's shape, particularly for objects with irregular geometries. As shown in Fig. 1 (a),

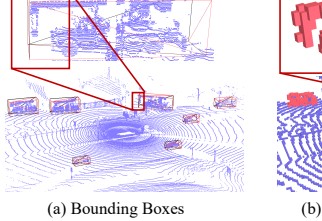 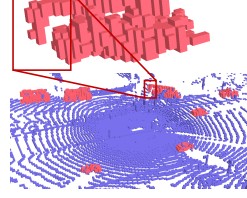

(a) Bounding Boxes          (b) Scene-Level Occupancy

Figure 1: Bounding Box vs. Occupancy. Occupancy can better represent the crane's shape than the bounding box.

---

*Works done during the internship at TuSimple.

38th Conference on Neural Information Processing Systems (NeurIPS 2024).

the crane is perfectly enclosed by a 3D bounding box. However, its boom, which is a long protrusion relative to the cab, results in a significant amount of unoccupied space within the 3D bounding box. Nevertheless, algorithms that employ 3D bounding boxes as a perception result inherently assume that the space within the bbox is fully occupied, thereby deeming the space enclosed by the 3D bounding box as impassable. Consequently, when addressing complex and irregularly shaped objects, bounding boxes are inadequate in providing fine-grained perceptual outcomes, which can consequently impact the precision of subsequent tasks, such as planning and control.

Considering the limitation of 3D bounding boxes, occupancy representation has emerged as a promising alternative for 3D scene perception [30, 29, 33]. As shown in Fig. 1 (b), occupancy representation discretizes the 3D space into a volumetric grid, wherein each voxel is classified as *occupied* or *free*. Compared to 3D bboxes, this representation more effectively captures irregular shapes, thereby enhancing accurate planning and control. Real-time scene-level occupancy generation from sensor inputs is non-trivial, presenting challenges not only for vision-centric inputs due to the absence of depth sensing, but also for LiDAR sensors because of the sparsity of each LiDAR scan (see Fig. 2 (b)). Thus,

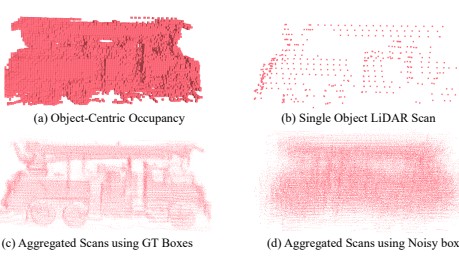

(a) Object-Centric Occupancy  (b) Single Object LiDAR Scan

(c) Aggregated Scans using GT Boxes  (d) Aggregated Scans using Noisy boxes

Figure 2: Generating occupancy from LiDAR scans is non-trivial for foreground objects due to sparsity and detection drifts.

existing approaches [35, 36] leverage neural networks to predict occupancy in a data-driven manner. Due to computational constraints, these methods typically produce low-resolution occupancy grids for large scene perception (*e.g.*, $200 \times 200 \times 16$ with a voxel size of $(0.4m)^3$ in [29]) or requires intensive training for implicit representation [11, 17], which remains insufficient and inefficiency for practical use.

Another feasible way to build occupancy grids is directly voxelizing the LiDAR point cloud. To alleviate the sparsity problem (Fig. 2 (b)), aggregating multiple LiDAR scans is an effective way for background. However, for foreground objects, the occupancy construction becomes challenging as it requires accurate detection and tracking to compensate for their potential movements. In real-time applications, 3D detection is prone to drift, and tracking algorithms may lose or mismatch objects, resulting in inaccurate tracklets. As illustrated in Fig. 2(d), directly aggregating point clouds from inaccurate tracklets can lead to extremely blurry shape representations. Such inaccuracies accumulate over time, progressively degrading the reliability of the shape representation.

Based on these observations, we introduce **object-centric occupancy** as a supplement to object bounding boxes, providing a more detailed structural description for objects' intrinsic geometry. In contrast to its scene-level counterpart, object-centric occupancy exclusively focuses on foreground objects, allowing for higher voxel resolutions even in large scenes. To encourage the advancement of object-centric occupancy perception, we present a novel object-centric occupancy dataset, which is constructed from scratch using an automated pipeline. We then propose a robust sequence-based occupancy completion network. By aggregating temporal information from history observations using attention, our network effectively handles detection drifts and accurately predicts the complete object-centric occupancy. Furthermore, our method employs an implicit shape decoder to generate dynamic-size occupancy and reduce training costs through queries on selective position. Our experiments under Waymo Open Dataset (WOD) [27] reveal that our method exhibits robust performance in completing object shapes even under noisy detection and tracking conditions. With the implicit shape descriptor, we demonstrate that performance of state-of-the-art 3D object detectors can also be improved, particularly for incomplete or distant objects.

## 2 Related Work

### 2.1 3D Occupancy Prediction and Shape Completion

3D semantic occupancy prediction (SOP) [16, 29, 34, 30, 12] has become a critical task in vision-centric autonomous driving, where algorithms primarily perceive the environment using RGB cameras. These vision-centric models typically discretize the surrounding environment into a volumetric grid

and predict the occupancy status of each voxel by properly aggregating information from single-/multi-view RGB image(s). For occupied voxels, the models additionally predict the corresponding semantic class. Another similar task is 3D semantic scene completion (SSC) [26]. Unlike SOP, which only needs to predict the occupancy for visible regions, SSC additionally requires the model to determine the occupancy status at unseen regions. It is worth noting that although SOP and SSC are predominantly associated with vision-centric approaches, they are also applicable to sparse LiDAR or multi-modal inputs [35, 36]. Existing SOP and SSC methods primarily focus on scene-level occupancy, while our work concentrates on object-centric occupancy for better shape representation. Besides, semantics for occupied voxels are not necessary for our setup, as our primary concern is the geometric structure within an object bbox, whose class label is given. Unlike our occupancy-based method, a majority of shape completion approaches focus on surface reconstruction of objects [1, 23]. However, surface-based representations are less suitable for autonomous driving perception, as they do not directly support tasks like collision avoidance.

## 2.2 3D Object Detection with Long Sequences

As demonstrated in [38, 41, 24], a single-frame detector can directly benefit from temporal information by taking the concatenation of several history frames as inputs. Although such a simple multi-frame strategy shows noticeable improvements, the performance becomes easily saturated as the number of input frames increases (*e.g.*,$2 \sim 4$ frames). Besides, the computational cost grows significantly as the number of input frames increases, which is not ideal for real-time applications. To remedy this issue, [9] employs a residual point probing strategy to remove redundant points in the multi-frame inputs. Besides, [3] opts for an object-centric approach that conducts the temporal aggregation at the level of tracklet proposals, which allows for longer sequences (*i.e.*,16 frames) to be processed with lower computational costs. Furthermore, [21, 8] demonstrate human-level detection performance by leveraging past and future information of entire object tracks. However, they are limited to offline applications since they require access to future frames. More recently, MoDAR[14] improves detection by augmenting LiDAR point clouds using motion forecasting outputs, which consist of future trajectory points predicted from long history subsequences (*i.e.*, 90 frames). Compared to MoDAR[14], our method is able to aggregate all the historical information via the compact implicit latent embeddings. Besides, our method goes beyond detection by predicting the complete object-centric occupancy for each proposal.

## 2.3 Implicit Neural Representation

Implicit shape representation [13] represents 3D shapes with a continuous function. Compared to traditional explicit representations (*e.g.*, point clouds, meshes, volumetric grids), implicit representations can describe shape structure in continuous space, and are more memory-efficient. Rather than manually designing the implicit function, recent works [18, 19, 39, 22] propose to learn the implicit function from data. Specifically, they employ neural networks to approximate the implicit function, which can be trained in a data-driven manner. These neural functions typically take continuous 3D coordinates as inputs and output the related shape attributes at the queried positions (*e.g.*, color, density, signed distance, etc.) For example, [19] learns a signed distance function (SDF) from high-quality 3D meshes for better shape representation. While [18] learns a neural radiance field from multi-view images to achieve better view synthesis. Our implicit shape decoder shares similarities with DeepSDF introduced in [19]. However, instead of predicting the signed distance at a queried position, we predict its occupancy probability.

## 3 Object-Centric Occupancy Dataset

High-quality datasets are critical for learning-based methods. However, existing datasets do not satisfy our requirements for object-centric occupancy perception due to unaligned coordinate systems and reduced resolutions. We discuss these limitations and introduce our automated annotation pipeline in the following subsections.

### 3.1 Object-Centric vs. Scene-Level Occupancy

Occupancy representation discretizes the 3D space into a volumetric grid, wherein each voxel is classified as *occupied* or *free*. Given our objective is to more accurately represent complex *object* structures, background elements — despite their extensive coverage — are not our primary focus. Therefore, we define object-centric occupancy as a 3D grid centered on the object's coordinate. As different object instances vary in size, their corresponding occupancy resolutions also vary, if given a predefined voxel size. In contrast, existing scene-level occupancy datasets [29, 34, 33] use an occupancy volume to represent an entire scene centered at the ego vehicle's coordinate system. Since all scenes are bounded by a fixed range, the occupancy resolution remains constant when the voxel size is given.

One convenient way to construct our object-centric occupancy dataset is to extract object occupancy from existing ego-centric datasets using object detection annotations. However, this approach has two significant limitations. Firstly, as scene-level occupancy is centered at the ego vehicle's coordinate system, the extracted object voxels may appear jagged due to coordinate misalignments, as shown in Fig. 3. Transforming these jagged object voxels to the object's coordinate system inevitably leads to information loss. Secondly, existing scene-level datasets have adopted a large voxel size (*e.g.*, $(0.4m)^3$) to save computational costs for large scenes. However, this voxel size is inadequate for capturing the fine-grained details of objects, especially for smaller objects. For this reason, we introduce an automated pipeline to annotate the object-centric occupancy dataset from scratch.

### 3.2 Dataset Generation Pipeline

Similar to previous scene-level approaches [29, 33], we can construct object-centric occupancy annotations based on any existing 3D detection datasets [27, 2]. However, instead of generating an occupancy volume for the entire scene, we create it for each annotated object instance under its local coordinate system.

For each designated object, we gather points within its annotated bounding boxes over time, transform these points from sensor coordinates to the bounding box coordinates and aggregate them into a dense point cloud. After that, we directly voxelize it under the local object coordinate system, yielding the object-centric occupancy grid.

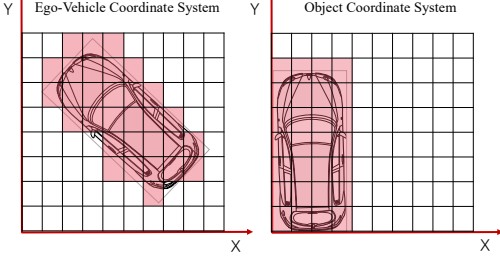

Figure 3: Occupancy grids defined in the ego-vehicle (left) and object-centric (right) coordinate systems. The object shape is jagged in the ego-vehicle occupancy grid due to coordinate misalignment.

Additionally, we perform occlusion reasoning to classify unoccupied voxels as either free or unobserved by comparing each voxel center's range value to raw range images from LiDAR scans. This strategy is significantly faster than traditional ray-casting [29]. After finishing the annotation, every tracked object within the detection dataset is associated with an object-centric occupancy grid. This grid is centered at the local coordinate and has a size determined by the object's size and the desired resolution. Please refer to Appendix A.1 for more details about the dataset generation pipeline.

## 4 Sequence-based Occupancy Completion Network

Fig. 4 illustrates the architecture of our object-centric occupancy completion network. Our method utilizes an object sequence as input, formulated as a $\{(\mathcal{P}_t, \mathcal{B}_t)\}_{t=0}^T$, where $\mathcal{P}_i \in \mathbb{R}^{N \times 3}$ is the point cloud at timestamp $t$ and $\mathcal{B}_t \in \mathbb{R}^7$ is the corresponding noisy 3D object bbox. The input sequence can be generated using off-the-shelf 3D detection [38, 6] and tracking [32] systems. Our main objective is to predict the complete object-centric occupancy grid for each proposal in the trajectory. Additionally, we use the occupancy features to further refine the detection results of the 3D detector.

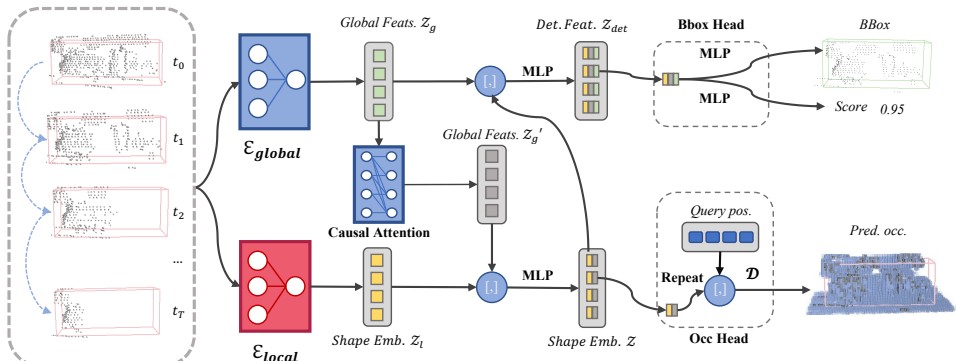

Figure 4: Architecture overview. The network takes a noisy object sequence as input and outputs the complete object-centric occupancy volume and refined bounding box for each proposal. The notation [,] denotes the concatenation operation. 'global'/'local' indicates features from global/local coordinate system.

## 4.1 Dynamic-Size Occupancy Generation via Implicit Decoding

Our network primarily focuses on Regions of Interest (RoIs) defined by object proposals. Given that different objects have varying sizes and proposals for the same object may also vary due to inaccurate detection, efficiently decoding the occupancy volume from feature space for each dynamic-sized proposal poses a significant challenge. Conventional scene-level occupancy perception approaches [30, 34] typically apply dense convolution layers to decode the occupancy volume. However, this strategy encounters several limitations in the context of dynamic-size object-centric occupancy. First, since we require feature interaction across timestamps, the features for different proposals are better if in the same size. However, decoding a dynamic-sized volume from a fixed-size feature map is non-trivial for convolution. Secondly, the dense convolution operation becomes computationally expensive for high occupancy resolution. One alternative is sparse convolution [5, 10], however, it cannot fill the unoccupied voxels with the correct occupancy status.

Drawing inspiration from the recent success of implicit shape representations [18, 19], we tackle the aforementioned challenge through an **implicit** shape decoder $\mathcal{D}$. This decoder is capable of predicting the occupancy status of any position within the RoI based on its corresponding latent embedding. Specifically, the decoder takes in the latent embedding $z$ along with a query position $q \in \mathbb{R}^3$ at the RoI coordinate, and subsequently outputs the occupancy probability at $q$:

$$p = \mathcal{D}(z, q), \tag{1}$$

where $\mathcal{D} : \mathbb{R}^e \times \mathbb{R}^3 \mapsto \mathbb{R}_{[0,1]}$ is implemented as an MLP. The latent $z \in \mathbb{R}^e$ is a fixed-length embedding depicting the geometrics within the RoI. The latent $z$ and query position $q$ are concatenated before being sent to $\mathcal{D}$. Besides enabling flexible feature interaction and efficient computation, the implicit shape decoder also allows for easier occupancy interpolation or extrapolation with continuous query positions.

## 4.2 Dual Branch RoI Encoding

Having the implicit shape decoder in place, the next step is to obtain a latent embedding $z$ that accurately represents the complete object shape within the RoI. To achieve accurate shape completion and detection, two information sources are essential: 1) the partial geometric structure of each RoI, and 2) the motion information of the object over time. To make different RoIs share the same embedding space, we encode each RoI under a canonical local coordinate system. However, transforming the RoI to the local coordinate system inevitably loses the global motion dynamics of the object, reducing the network's ability to handle detection drifts. Therefore, we encode each RoI using two separate encoders: $\mathcal{E}_{\text{local}}$ that encodes the RoI in the local coordinate system and $\mathcal{E}_{\text{global}}$ in the global coordinate system.

Specifically, we employ the sparse instance recognition (SIR) module in FSD[6] as our RoI encoder. SIR is a PointNet-based network [20] characterized by multiple per-point MLPs and max-pooling layers. Drawing inspiration from LiDAR R-CNN [15], we additionally enhance the point cloud with size information of the RoI. This augmentation involves decorating each point within the RoI

with its offset relative to the boundary of the RoI, enabling it to be box-aware. All points are transformed to the local coordinate system defined by the detected bounding box before being sent to $\mathcal{E}_{\text{local}}$. Conversely, $\mathcal{E}_{\text{global}}$ directly encodes the RoI in the global coordinate system. For a given object sequence $\{(\mathcal{P}_t, \mathcal{B}_t)\}_{t=0}^{T}$, we separately encode each RoI using $\mathcal{E}_{\text{local}}$ and $\mathcal{E}_{\text{global}}$, yielding two sets of latent embeddings $\mathcal{Z}_l$ and $\mathcal{Z}_g \in \mathbb{R}^{T \times e}$.

### 4.3 Feature Enhancement via Temporal Aggregation

After RoI encoding, we use the motion information from $\mathcal{Z}_g$ to enrich the local shape latent embeddings $\mathcal{Z}_l$. First, we employ a transformer mechanism [31] to $\mathcal{Z}_g$ to enable feature interaction across timestamps. To ensure online applications, we restrict each RoI feature in $\mathcal{Z}_g$ to only attend to its historical features, thereby preventing information leakage from future timestamps:

$$\mathcal{Z}_g' = \text{CausalAttn}(\mathcal{Z}_g + \gamma(\mathcal{T}) + \phi(\mathcal{B})), \tag{2}$$

where CausualAttn is a causal transformer that restricts the attention to the past timestamps. $\gamma(\cdot)$ is a sinusoidal positional encoding [31] that encodes the temporal timestamp $\mathcal{T} \in \mathbb{R}^{T \times 1}$. $\phi(\cdot)$ is a learnable MLP that encodes the bbox information $\mathcal{B} \in \mathbb{R}^{T \times 7}$ in the global coordinate system.

Next, we fuse the enriched global latents $\mathcal{Z}_g'$ with the local latents $\mathcal{Z}_l$ to obtain the final latent embeddings $\mathcal{Z} \in \mathbb{R}^{T \times e}$:

$$\mathcal{Z} = \text{MLP}(\text{Concat}(\mathcal{Z}_l, \mathcal{Z}_g')), \tag{3}$$

where 'Concat' denotes the concatenation operation, and 'MLP' is a multi-layer perceptron that projects the concatenated features to the desired dimension $c$.

### 4.4 Occupancy Completion and Detection Refinement

Given the final latent embeddings $\mathcal{Z}$, we can predict the complete object-centric occupancy volume for each proposal by querying the implicit shape decoder $\mathcal{D}$ at different positions. During training, we randomly sample a fixed number of query positions within each RoI to compute the loss. During inference, we query the decoder at all voxel centers within the RoI to obtain the complete occupancy volume. Since $\mathcal{Z}$ now encodes information of complete object shapes, it provides more geometric information for better detection. To retain motion information, we additionally fuse $\mathcal{Z}$ with the global RoI feature $\mathcal{Z}_g$:

$$\mathcal{Z}_{\text{det}} = \text{MLP}(\text{Concat}(\mathcal{Z}, \mathcal{Z}_g)). \tag{4}$$

The fused feature $\mathcal{Z}_{\text{det}}$ is then fed into a detection head for bbox and score refinement (Fig. 4).

### 4.5 Loss functions

The overall training loss consists of three components: the occupancy completion loss $\mathcal{L}_{\text{occ}}$, the bbox loss $\mathcal{L}_{\text{det}}$, and the objectness loss $\mathcal{L}_{\text{score}}$:

$$\mathcal{L} = \mathcal{L}_{\text{occ}} + \lambda_{\text{det}}\mathcal{L}_{\text{det}} + \lambda_{\text{score}}\mathcal{L}_{\text{score}}, \tag{5}$$

where $\lambda_{\text{det}} = 2$ and $\lambda_{\text{score}} = 1$ are hyperparameters that balance the three losses. We use the binary cross-entropy loss for $\mathcal{L}_{\text{occ}}$ and $\mathcal{L}_{\text{score}}$, and the L1 loss for $\mathcal{L}_{\text{det}}$.

## 5 Experiments

### 5.1 Implementation Details

**Position Query Sampling.** During training, we randomly sample 1024 voxel centers and corresponding occupancy statuses from each annotated occupancy as the position queries. To ensure the occupancy prediction is not biased, we adopt a balanced sampling strategy, where 512 points are sampled from the occupied voxels and 512 from the free voxels. For an RoI that matches a ground-truth (GT) bbox, we transform the corresponding query set to its coordinate system using the relative pose between the RoI and the bbox. These position queries are then sent to the implicit

decoder $\mathcal{D}$ to compute the occupancy loss. During the inference, we generate the dense occupancy volume for each RoI by querying the decoder at all voxel centers within the RoI under the local coordinate system.

**Network Training.** In order to generate inputs for our network, we first use FSD [6] and Center-Point [38] as our base detectors to generate object proposals. Then we leverage ImmortalTracker [32] to associate the detection results into object tracklet proposals. We use the generated object tracklet proposals in addition to GT tracklets as our training sequences. To facilitate parallel training, we regularize each tracklet to a fixed length of 32 frames via padding or cutting during training. To achieve faster convergence, we compute the loss at all timestamps within each tracklet instead of only at the last one. During the inference, the model outputs the refined box at timestamp $t$ by looking at all the history boxes.

## 5.2 Dataset and Evaluation Metrics

**Dataset.** Our method is evaluated on the Waymo Open Dataset (WOD)[27]. We use the official training set, comprising 798 sequences for training, and 202 sequences for evaluation. We apply our automatic pipeline on WOD to construct the object-centric occupancy annotations with the voxel size set to 0.2m. All experiments are conducted on rigid objects (*i.e.*, vehicles) to ensure accurate evaluation of shape completion using our annotated ground-truths.

**Evaluation Metrics.** For shape completion, we adopt the widely-used intersection-over-union (IoU) to evaluate the quality of the predicted occupancy volumes. Due to the object-centric nature of our method, we cannot calculate the IoU directly between the predicted and the ground-truth occupancy volumes because they are in different coordinate systems and may have different sizes (noisy RoI vs. GT box). To overcome this issue, we employ a two-step process as illustrated in Fig. 5. Firstly, we transform the ground-truth (GT) box to the coordinate system of the RoI using the relative pose. This

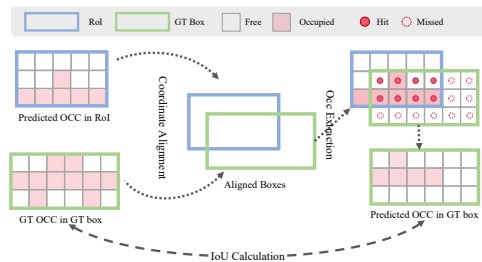

Figure 5: Illustration for occupancy evaluation.

transformation aligns the GT box with the RoI, enabling a consistent comparison. Subsequently, we determine the predicted occupancy status of each voxel center within the transformed GT box. For voxels falling inside the RoI (hit), their occupancies are determined by the corresponding predicted occupancies within the RoI. On the other hand, voxels located outside the RoI (missed) are considered as free. By applying this process, we construct a predicted occupancy volume within the GT box. Finally, we compute the IoU by comparing the predicted occupancy volume in the GT box with the ground-truth occupancy volumes. During the IoU calculation, we ignore unobserved voxels in the GT volume for a fair assessment. Besides, RoIs that do not intersect with any GT boxes are excluded from the evaluation. We also report mean IoU that is respectively averaged at track and box levels to provide a more detailed evaluation.

For object detection, we adopt the official 3D detection metrics in WOD [27], including Average Precision (AP) and Average Precision Weighted by Heading (APH) at IoU thresholds of 0.7 for vehicles. Meanwhile, based on the number of points contained within each object, the data is divided into two difficulty levels: LEVEL 1, where the number of points is greater than 5, and LEVEL 2, where the number of points is between 1 and 5.

## 5.3 Shape Completion Results

**Comparison against Baseline.** Since object-centric occupancy is a novel task, no learning-based methods can be used for comparison as far as we are acknowledged. We compare our method with the baseline that directly accumulates and voxelizes the history point clouds within the noisy tracklet proposals. We evaluate the shape completion performance on three types of tracklet inputs: ground-truth (GT) tracklets, tracklets generated by CenterPoint (CP) [38], and tracklets generated by FSD [6]. As shown in Tab. 1, the shape completion performance is strongly correlated with the quality of the input tracklets, where better tracklets lead to better shape completion. In all cases, our method outperforms the baseline, even when the input tracklets are noise-free GTs. This is because our method can effec-

tively complete the object shape even at early timestamps by leveraging learned knowledge from training data, whereas the baseline only becomes effective at later timestamps when more views are visible.

**Robustness.** To simulate unsatisfied detection and tracking results, we add some slight noise to GT box proposals. From Tab. 1 we can find that the baseline performance drops significantly (>10% IoU), while our method maintains a stable performance in this case (<5% IoU), demonstrating the robustness of our method to noisy inputs. Compared to noisy GT tracklets, tracklets generated by CP and FSD may additionally contain mismatched or missed targets, leading to a more significant performance drop from the baseline. In contrast, our method demonstrates its strong robustness to these noisy and inaccurate tracklets. These results indicates that our method can effectively complete the object shape even when the input tracklets are noisy or inaccurate.

| Tracklet Inputs | Method | IoU % | mIoU (track) % | mIoU (box) % |
|---|---|---|---|---|
| GT track | Baseline | 61.35 | 62.19 | 63.46 |
| | Ours | **69.15** | **64.05** | **67.91** |
| GT track + noise | Baseline | 50.39 | 45.21 | 48.59 |
| | Ours | **64.92** | **60.70** | **63.78** |
| | Ours-E | *69.30* | *64.11* | *68.04* |
| FSD track | Baseline | 44.28 | 34.77 | 42.61 |
| | Ours | **62.84** | **54.12** | **61.58** |
| | Ours-E | *68.38* | *60.96* | *67.22* |
| CP track | Baseline | 40.45 | 26.69 | 37.29 |
| | Ours | **57.99** | **44.94** | **55.10** |
| | Ours-E | *65.80* | *56.81* | *64.29* |
| FSDv2 track | Ours (no train) | 61.41 | 47.69 | 60.76 |

Table 1: Shape completion results on WOD val set. "-E" denotes using GT bbox which may outside the predicted RoIs.

**Results with GT bbox.** Thanks to the implicit shape decoder, our method has the potential to predict the occupancy status at any position even "outside" the RoI, which is non-trivial for the baseline or CNN-based methods. To demonstrate this ability, we conduct an experiment by querying the implicit decoder at all voxel centers within the GT box (even for those outside the RoI). Unlike our standard evaluation, where we simply treat the missed positions as free (see Fig. 5), we query the implicit decoder at these positions to obtain the predicted occupancy status. As shown in Tab. 1, the shape completion performance is further improved when considering the extrapolated results outside the RoIs (Ours-E), demonstrating the flexibility of our implicit shape representation.

**Generalization.** The last row in Tab. 1 presents occupancy completion results obtained by directly applying our trained model to the tracklet proposals generated by FSDv2 [7]. Due to better detection, our method with FSDv2 still outperforms the version with CenterPoint even without retraining. However, it performs slightly worse compared to using FSD tracklets, despite FSDv2 having better detection results than FSD. This indicates that significant detection improvements generally lead to better shape completion (FSDv2 vs. CenterPoint). However, for detectors with similar performance (e.g., FSD vs. FSDv2), improved detections do not necessarily guarantee better shape completion without retraining.

## 5.4 Object Detection Results.

**Main Results** Tab. 2 presents the 3D detection results on the WOD validation set. Significant improvements are observed when applying our methods to the tracklet proposals generated by CenterPoint [38] and FSD [6]. Compared to the previous state-of-the-art MoDAR [14], our method achieves notably greater enhancements on 1-frame CenterPoint (*e.g.*, 8.6% vs. 3.2% improvement in L1 AP). Applying our method to a more advanced detector, 1-frame FSD [6], still results in a noticeable improvement. This enhancement is more significant compared to adding MoDAR to a detector with similar performance (i.e., 3-frame SWFormer [28]). Furthermore, we achieve new state-of-the-art online detection results by applying our method to 7-frame FSD, attaining 83.3% AP and 75.7% APH on L1 and L2, respectively. This indicates our method's effectiveness in aggregating long-sequence information for object detection in addition to shape completion. Moreover, our method can be seamlessly integrated with other state-of-the-art detectors without requiring retraining on their respective tracklets in the training data. For example, applying our method (trained on CP and FSD tracklets) to FSDv2 [7] yields significant improvements, showcasing the strong generalization capability of our approach.

**Range Breakdown**. Distant objects are more challenging to detect due to their sparsity. We further analyze the detection performance across different distance ranges. As shown in Tab. 3, our improvements over the base detector become more pronounced as the distance

| Model | [0,30) | [30,50) | [50,+inf) |
|---|---|---|---|
| FSD [6] | 90.97 | 70.87 | 46.04 |
| + Ours | 92.55 (+1.58) | 75.83 (+4.96) | 53.85 (**+7.81**) |
| CenterPoint [38] | 89.26 | 65.72 | 37.53 |
| + Ours | 92.33 (+3.07) | 74.88 (+9.06) | 51.47 (**+13.94**) |

Table 3: Range breakdown (L2 mAP).

| Method | Frame [-p,+f] | Vehicle L1 3D | | Vehicle L2 3D | |
|--------|---------------|-----|------|-----|------|
| | | AP | APH | AP | APH |
| 3D-MAN [37] | [-15, 0] | 74.5 | 74.0 | 67.6 | 67.1 |
| CenterFormer [41] | [-3, 0] | 78.1 | 77.6 | 73.4 | 72.9 |
| CenterFormer [41] | [-7, 0] | 78.8 | 78.3 | 74.3 | 73.8 |
| MPPNet [3] | [ -3, 0] | 81.5 | 81.1 | 74.1 | 73.6 |
| MPPNet [3] | [-15, 0] | 82.7 | 82.3 | 75.4 | 75.0 |
| FSD++ [9] | [ -6, 0] | 81.4 | 80.9 | 73.3 | 72.9 |
| MVF++ [21] | [ -4, 0] | 79.7 | - | - | - |
| VoxelNeXt [4] | [ 0, 0] | 78.2 | 77.7 | 69.9 | 69.4 |
| HEDNet [40] | [ 0, 0] | 81.1 | 80.6 | 73.2 | 72.7 |
| CenterPoint* [38] | [ 0, 0] | 72.9 | 72.3 | 64.7 | 64.2 |
| +MoDAR [14] | [-91, 0] | 76.1 (+3.2) | 75.6 (+3.3) | 68.9 (+4.2) | 68.4 (+4.2) |
| CenterPoint‡ [38] | [ 0, 0] | 73.2 | 72.7 | 65.2 | 64.6 |
| +Ours | $[-\infty, 0]$ | 81.8 **(+8.6)** | 81.3 **(+8.6)** | 73.6 **(+8.4)** | 73.2 **(+8.6)** |
| SWFormer* [28] | [ 0, 0] | 77.0 | 76.5 | 68.3 | 67.9 |
| +MoDAR [14] | [-91, 0] | 80.6 (+3.6) | 80.1 (+3.6) | 72.8 (+4.5) | 72.3 (+4.4) |
| SWFormer* [28] | [ -2, 0] | 78.5 | 78.1 | 70.1 | 69.7 |
| +MoDAR [14] | [-91, 0] | 81.0 (+2.5) | 80.5 (+2.4) | 73.4 (+3.3) | 72.9 (+3.2) |
| FSD‡ [6] | [ 0, 0] | 78.7 | 78.3 | 70.1 | 69.7 |
| +Ours | $[-\infty, 0]$ | 82.8 **(+4.1)** | 82.3 **(+4.0)** | 74.8 **(+4.7)** | 74.4 **(+4.7)** |
| FSD‡ [6] | [ -6, 0] | 80.9 | 80.5 | 73.1 | 72.7 |
| +Ours | $[-\infty, 0]$ | **83.3** (+2.4) | **82.9** (+2.4) | **75.7** (+2.6) | **75.2** (+2.5) |
| FSDv2 [7] | [ 0, 0] | 79.8 | 79.3 | 71.4 | 71.0 |
| +Ours(no train) | $[-\infty, 0]$ | **83.2** (+3.4) | **82.7** (+3.4) | **75.2** (+3.8) | **74.7** (+3.7) |

Table 2: Detection results on WOD val set. *: reported by MoDAR [14] . ‡ : our re-implementation. The Frame column illustrates the indices of the frames that are used. Blue indicates the improvement over the baseline.

increases. This indicates that our method effectively addresses the sparsity issue in distant objects via shape completion.

## 5.5 Model Analysis.

In this section, we evaluate different design choices in our method and analyze their impact on the shape completion and detection performance. All the results are based on 1-frame FSD [6] tracklets.

**Single Branch vs. Dual Branch.** We first evaluate the performance when using only a single branch for RoI encoding. In this setting, only a local encoder $\mathcal{E}_{\text{local}}$ is used to encode the RoI in the local coordinate system. The encoded features are enhanced by the causal transformer and then used to generate occupancy and detection outputs. As shown in Tab. 4, the single-branch model is inferior to our dual-branch model in both shape completion and detection. This indicates that the motion information from the global branch is essential for accurate shape completion and detection refinement.

**Explicit vs. Implicit.** We then attempt to refine detection results using the explicit occupancy predictions. Specifically, we sample occupied voxel centers from each predicted occupancy volume and apply the RoI encoder $\mathcal{E}_{\text{global}}$ to generate the final feature used for detection (more details are in the Appendix A.2). However, as demonstrated in Tab. 4, this strategy leads to a significant performance drop. Due to the non-differentiable nature of the occupancy sampling process, the detection errors cannot be back-propagated to other components when relying on explicit occupancy predictions, resulting in unstable training. In contrast, our implicit shape representation allows for joint end-to-end training of shape completion and detection, leading to better performance.

| Model | IoU | Vehicle 3D AP/APH | |
|-------|-----|-----|-----|
| | | L1 | L2 |
| Ours | **62.84** | **82.80/82.31** | **74.83/74.36** |
| Single-Branch | 62.13 | 80.51/80.05 | 72.26/71.82 |
| Explicit Occ. | 61.50 | 80.20/79.71 | 71.93/71.48 |
| No Occ. Dec. | - | 81.10/80.40 | 73.00/72.30 |

Table 4: Analysis of different designs.

**Occupancy Helps Detection.** Finally, we evaluate the impact of the occupancy task on detection performance. We removed the OCC head from our full model and retrained it using only the detection loss. As shown in the last row of Tab. 4, the absence of the occupancy decoder results in a noticeable decline in detection performance. This suggests that the occupancy completion task not only explicitly enriches the object shape representation but also enhances detection by contributing additional geometric information to the latent space.

**Training & Testing Length.** Tab. 5 shows how the sequence lengths affect the performance of our method. We retrain our method using 8-frame and 16-frame tracklets, respectively. As indicated in the first 3 rows in Tab. 5, using longer sequences for training leads to better results. However, the performance improvement diminishes as the sequence length doubles. To strike a balance between performance and computational cost, we set our default training length to 32. Even trained with 32-frame tracklets, our method is flexible to handle various-length tracklets during inference. By default, we leverage all history frames to generate predictions at each timestamp. However, we can also generate predictions using a subset of historical

| Training Frames | Testing Frames | IoU | Vehicle 3D AP/ APH | |
| --- | --- | --- | --- | --- |
| | | | L1 | L2 |
| 32 | [-∞, 0] | **62.84** | **82.80/82.31** | **74.83/74.36** |
| 8 | [-∞, 0] | 62.43 | 80.79/80.29 | 72.57/72.10 |
| 16 | [-∞, 0] | 62.61 | 82.24/81.73 | 74.22/73.73 |
| 32 | [-7, 0] | 62.28 | 80.92/80.43 | 72.73/72.27 |
| 32 | [-15, 0] | 62.47 | 81.36/80.87 | 73.26/72.80 |
| 32 | [-31, 0] | 62.66 | 81.85/81.36 | 73.81/73.35 |
| 32 | [-63, 0] | 62.80 | 82.28/81.79 | 74.30/73.83 |
| 32 | [-∞, ∞] | 63.03 | 84.03/83.51 | 76.17/75.68 |

Table 5: Results for various sequence lengths.

frames to reduce computational costs. As shown in Tab. 5, [-63,0] frames for inference achieves similar performance as using all history frames. Moreover, our method can also be extended to handle offline scenarios. When the transformer attends to all timestamps including those future ones, the performance improves further, as demonstrated in the last row of Tab. 5.

**Computational Efficiency.** Tab. 6 shows the time and GPU memory cost of the proposed shape decoder. Since object tracklets vary in length, our method's running time may also vary with different inputs. Additionally, the dimension of the decoded object-centric occupancy depends on the detected bounding box. To ensure fair testing of running time, we standardized the input length to 32 and set the number of decode queries to 4096. As demonstrated in Tab. 6, the shape decoder only introduces a slight increase in computational cost, demonstrating its efficiency.

| Model | Avg. Time | Avg. GPU mem. |
| --- | --- | --- |
| w/o shape decode | 4.08ms | 2499MB |
| w/ shape decoder | 4.23ms | 2565MB |

Table 6: Cost analysis of the shape decoder.

## Limitations

Technically speaking, our automatic occupancy annotation relies on the rigid-body assumption, which may not be accurate for deformable objects. Consequently, our experiments focus on vehicle objects since they are rigid. Although our method can be applied to other deformable object categories, accurate evaluation for deformable objects cannot be guaranteed due to considerable noise in the ground-truth data.

## Conclusion

In this work, we introduce a novel task, object-centric occupancy, which extends the traditional object bounding box representation to provide a more detailed description of the object shape. Compared to its scene-level counterpart, object-centric occupancy enables higher voxel resolution in large scenes by focusing on foreground objects. To facilitate object-centric occupancy learning, we build an object-centric occupancy dataset using LiDAR data and box annotations from the Waymo Open Dataset (WOD). We further propose a novel sequence-based occupancy completion network that learns from our dataset to complete object shapes from noisy object proposals. Our method achieves state-of-the-art performance on both shape completion and object detection tasks on WOD. We believe that our work will inspire future research in perception tasks in the context of autonomous driving.

## Acknowledgements

This work was supported by NSFC with Grant No. 62293482, by the Basic Research Project No. HZQB-KCZYZ-2021067 of Hetao Shenzhen HK S&T Cooperation Zone, by Shenzhen General Program No. JCYJ20220530143600001, by Shenzhen-Hong Kong Joint Funding No. SGDX20211123112401002, by the Shenzhen Outstanding Talents Training Fund 202002, by Guangdong Research Project No. 2017ZT07X152 and No. 2019CX01X104, by the Guangdong Provincial Key Laboratory of Future Networks of Intelligence (Grant No. 2022B1212010001), by the Guangdong Provincial Key Laboratory of Big Data Computing, CHUK-Shenzhen, by the NSFC 61931024&12326610, by the Key Area R&D Program of Guangdong Province with grant No. 2018B030338001, by the Shenzhen Key Laboratory of Big Data and Artificial Intelligence (Grant No. ZDSYS201707251409055), and by Tencent & Huawei Open Fund.

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

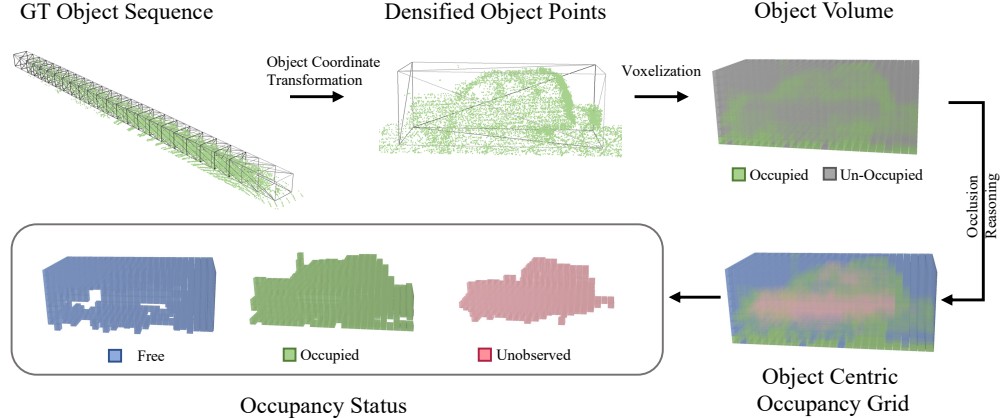

Figure 6: Our object-centric occupancy annotation pipeline.

# A   Appendix

## A.1   Dataset Generation Pipeline

Our annotation pipeline is illustrated in Fig. 6. Leveraging LiDAR scans and detection annotations from a base 3D dataset, our pipeline probes dense occupancy grids by aggregating multi-frame LiDAR point clouds and then executes occlusion reasoning to discriminate between free and unobserved voxels. Compared to ego-centric approaches [29, 33], our methodology primarily differs by focusing on annotated objects instead of the entire scene. For each designated object, we gather points within its annotated bounding boxes over time, transform these points from sensor coordinates to the bounding box coordinates and aggregate them into a dense point cloud. Unlike scene-level occupancy, we do not transform the densified object point cloud back to each ego-vehicle coordinate for occupancy construction. Instead, we directly voxelize it under the local object coordinate system, maintaining object-centric precision. While densified object point clouds encode better shape information than a single LiDAR scan, it's important to note that unoccupied voxels (grey voxels in Fig. 6) does not necessarily indicate free space; they may be unobserved by the LiDAR due to occlusion. Hence, an occlusion reasoning process is required to distinguish between voxels that are truly free and those that are unobserved. Basically, an unoccupied voxel is considered *free* if it is traversed trough by a LiDAR ray, and *unobserved* otherwise. Instead of the time-consuming ray-casting operation used in [29], we adopt a more efficient approach by leveraging range information from raw range images. Specifically, for each unoccupied voxel, we first convert its center to the range image format using sensor intrinsics and extrinsics at a specific timestamp $t$, yielding a 2D-pixel index $(u_t, v_t)$ and a range value $r_t$. Next, we decide its status by comparing its range with the original range image at timestamp $t$:

$$
\begin{aligned}
&if \quad r_t < \mathrm{R}_t[u_t, v_t]: &&\quad free \\
&else: &&\quad unobserved
\end{aligned}
\tag{6}
$$

where $\mathrm{R}_t \in \mathbb{R}^{H \times W}$ is the raw range image captured by the LiDAR sensor at timestamp $t$. We do this for all timestamps to decide the final status of the voxel. And a voxel is considered *unobserved* only if it is not traversed by any LiDAR ray at any timestamp.

Fig. 7 shows some examples of our object-centric occupancy annotations.

## A.2   Alternative Design Choices

Fig. 8 and Fig. 9 illustrate the pipelines of the single-branch model and the model using explicit occupancy for detection, respectively. The two global RoI encoders $\mathcal{E}_{\text{global}}$ in Fig. 9 share the same weights. We additionally add an extra channel to each point feature to indicate whether it is from raw point clouds or from the predicted occupancy volume.

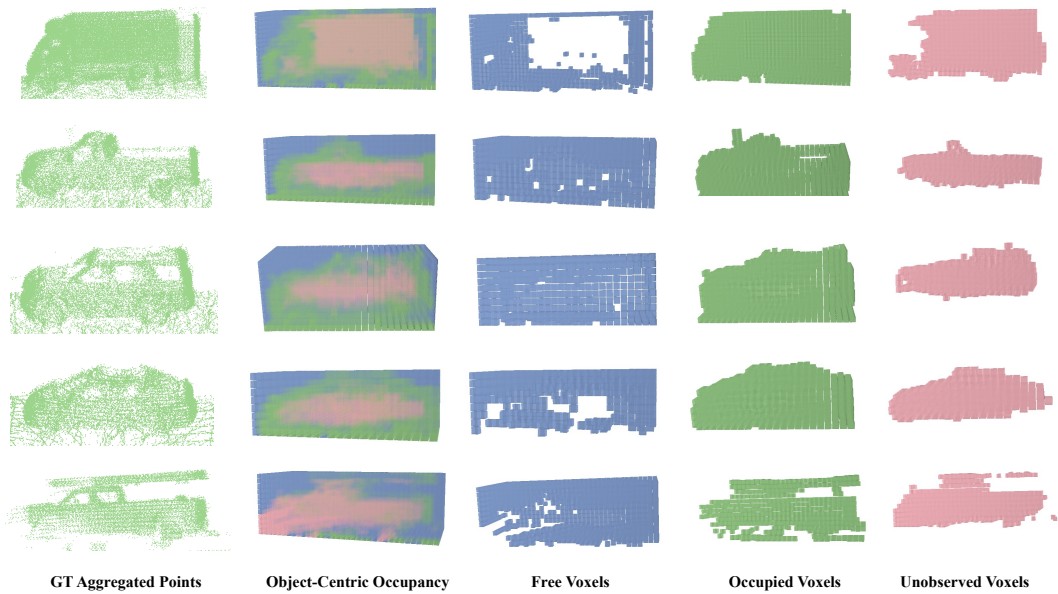

| GT Aggregated Points | Object-Centric Occupancy | Free Voxels | Occupied Voxels | Unobserved Voxels |

Figure 7: Visualization of our object-centric occupancy annotations. The first colume shows the GT-aggregated LiDAR points. The second column shows our annotated object-centric occupancy volume. The last three columns respectively show the occupancy at free, occupied and unobserved status.

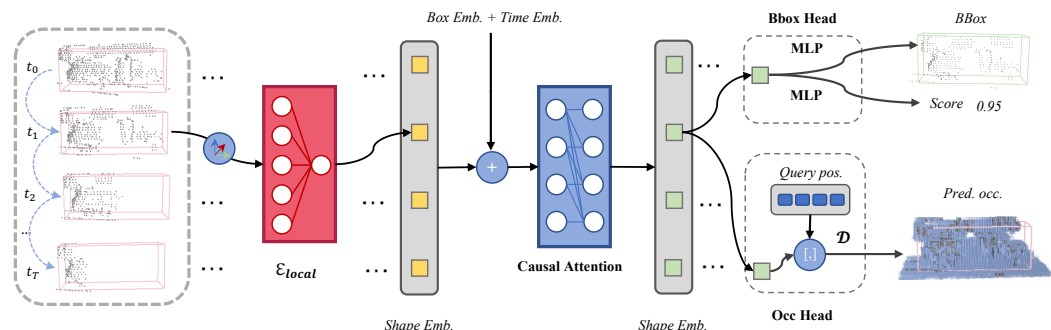

Figure 8: Single-branch model architecture.

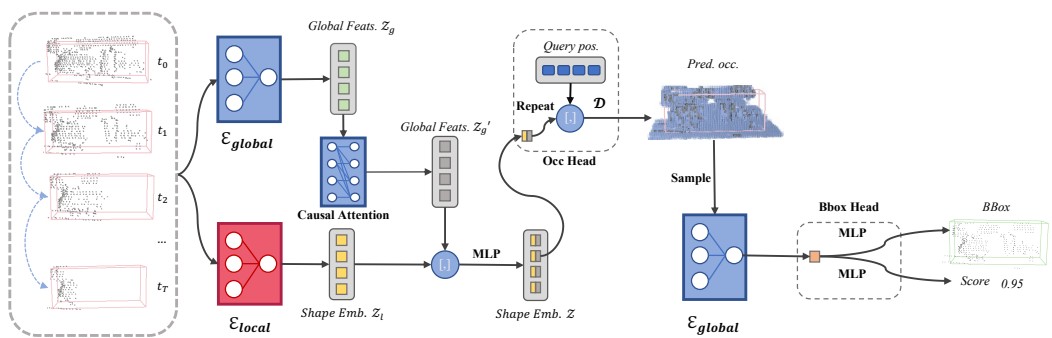

Figure 9: Architecture of using explicit occupancy for detection.

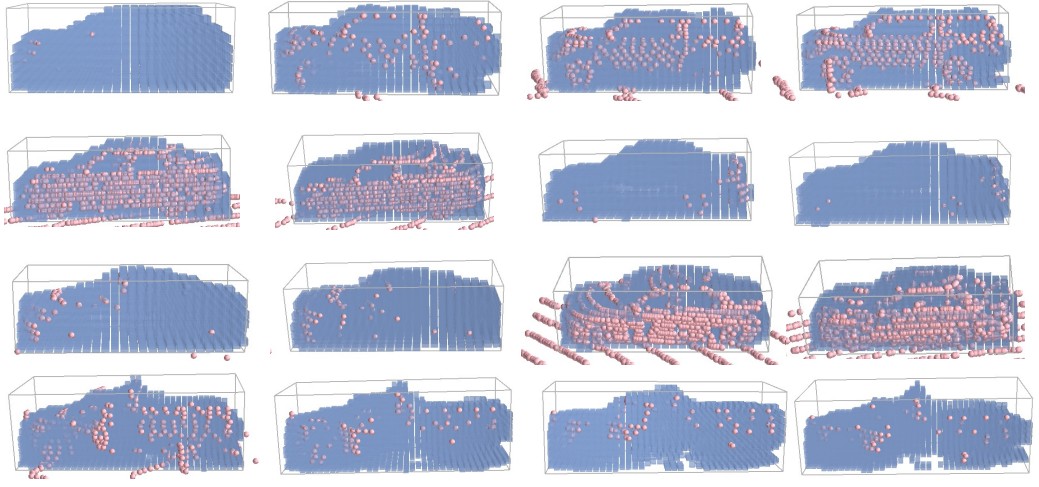

Timestamp

Figure 10: Visualization of the object-centric occupancy prediction. Different rows denote different object instances. Pink points indicate LiDAR points. Blue cubes represent the predicted occupied voxels.

### A.3 Training Details & Hyper-parameters

We train our model using the Adam optimizer with an initial learning rate of $1e$-4 and a batch size of 8. The model is trained for 24 epochs with the learning rate scheduled by the cosine annealing strategy. We use a transformer with 3 layers, 4 heads, and a hidden dimension of 512. The model is implemented using PyTorch and trained on 8 NVIDIA 3090 GPUs.

### A.4 Visualization of the Occupancy Prediction

Fig. 10 shows some examples of the occupancy prediction. Our method effectively predicts the object's shape even when it is extremely occluded. Additionally, our method effectively completes the object shape even at early timestamps, with shape completion improving as the sequence extends.

we've also included several surface renderings of the predicted occupancy in Fig. 11 and Fig. 12. These renderings were obtained by applying marching cubes to the decoded volumetric grids using a level of 0.5. The renderings demonstrate that our method can complete shapes even when the current point cloud is extremely sparse. Due to the use of 0.2m voxel size, the resolution of our predicted occupancy may not support high-quality rendering. For example, the resolution for a typical sedan (let's assume its dimensions are 4.5m* 1.8m * 1.4m) under our voxel size is 23 * 9 * 7. In contrast, common shape completion methods typically use a resolution of 128 x 128 x 128 or higher to facilitate high-quality rendering. It should be noted that for our purposes, high-quality rendering is not required. Although the selected voxel size of 0.2 meters may not provide highly detailed rendering, it is sufficient for downstream driving tasks and ensures computational affordability.

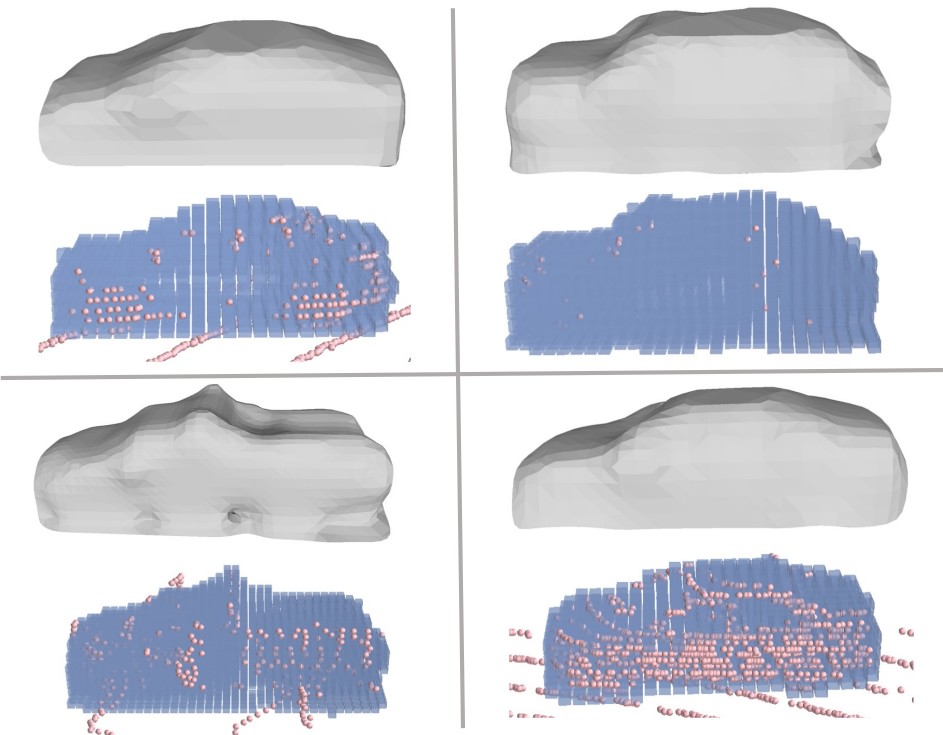

Figure 11: The renderings of predicted occupancy decoded from the shape codes for common vehicles. Top: extracted mesh from the occupancy using marching cube. Bottom: predicted occupancy and point cloud input.

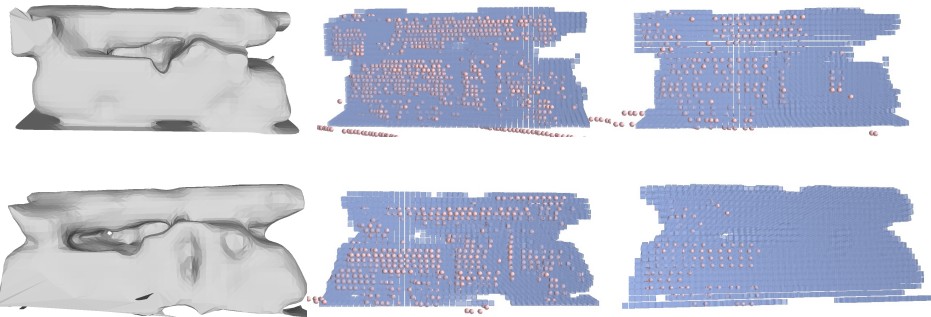

Figure 12: The renderings of complex vehicles. Each row shows the rendering, the corresponding predicted occupancy and input point cloud, and another predicted occupancy with fewer input points. These results demonstrate that the predicted object occupancy can better represent complex shape structures than bounding boxes.

