# OpenReview forum: "Towards Flexible 3D Perception: Object-Centric Occupancy Completion Augments 3D Object Detection"
_NeurIPS.cc/2024/Conference — NeurIPS 2024 poster_

### Official Review · Reviewer_nYQd · 2024-07-12

**Soundness:** 2
**Presentation:** 3
**Contribution:** 3
**Rating:** 6
**Confidence:** 4

**Summary:**

This paper presents a new task named object-centric occupancy completion as a fine-grained object representation to supplement the coarse-grained 3D bounding boxes. To accomplish this task, a new dataset, which annotates instance-level high-resolution occupancy, is created in an automated pipeline. This paper also introduces an implicit shape decoder to fuse multi-frame information, predict instance occupancy and refine 3D bounding boxes. Experiments on Waymo datasets above several baselines demonstrate the effectiveness of the proposed method on both occupancy prediction and 3D detection.

**Strengths:**

1.	This paper is well-written and organized.

2.	A novel task, occupancy augments 3D object detection, and a corresponding new instance-level occupancy datasets is proposed.

3.	A implicit shape decoder is proposed and achieves great improvements both in occupancy and 3D detection.

**Weaknesses:**

1.	The motivation of this paper does not seem to be very reasonable. The authors claim that a. high-resolution scene-level occupancy is constrained by computational cost and foreground objects is more import, and b. 3D detection is too coarse to capture the object geometry information. So why not just predict foreground instance-level occupancy in the whole scene, instead of pursuing higher detection accuracy by using the occupancy results?
2.	Time and memory cost bought by the proposed shape decoder are not provided. The paper is trying to make a trade-off between occupancy and detection in fine-/coarse-grained level and computational cost level. But the authors only report the occupancy and detection accuracy.
3.	Some methods, like VoxelNeXt, FSDv2, HEDNet are missing and are not compared in Table 1.
4.	Typos/mis-leading descriptions. For example, ‘Tab. 5.4’ on line 351 -> ‘Tab. 3’.

**Questions:**

1.	Why not just predict foreground instance-level occupancy in the whole scene, instead of pursuing higher detection accuracy by using the occupancy results? (the same as weakness 1)
2.	Could you provide the computational cost of your method or the proposed module?

**Limitations:**

The authors adequately addressed the limitations.

---

> ### Author Rebuttal · Authors · 2024-08-03
>
> 1. **Why not just predict foreground instance-level occupancy in the whole scene, instead of pursuing higher detection accuracy by using the occupancy results?**
>
>
>     1. To predict foreground instance-level occupancy for the entire scene, it is essential to distinguish the foreground from the background. However, separating the foreground from the background without detection results is a non-trivial task. Therefore, we leverage detection to obtain foreground objects.
>     2. Pursuing higher detection accuracy is not our main objective.  Our main goal is to obtain object-centric occupancy to provide a more flexible representation for downstream driving tasks. However, when we attempted to aggregate multiple frames of occupancy grids to make the shape more complete, we realized that these occupancy grids could also serve as an excellent medium for multiple frame information fusion. Thus, we designed a simple fusion network to examine the benefits for the detection task.
>
> 2. **Time and memory cost**
>
>     Since object tracklets vary in length, our method's running time may also vary with different inputs. Additionally, the dimension of the decoded object-centric occupancy depends on the detected bounding box. To ensure fair testing of running time, we standardized the input length to 32 and set the number of decode queries to 4096.
>
>     We conducted the inference with batch_size = 1 using standardized inputs on a single 3090 GPU and computed the average running costs. The results are presented in the table below. We can observe that using the shape decoder does not significantly affect the computational cost.
>
>     | model | Avg. Inference Time | Avg GPU memory Cost |
>     | --- | --- | --- |
>     | w/o shape decoder | 4.08ms | 2499MB |
>     | w/ shape decoder | 4.23ms |2565MB |
>
>
> 3. **Some methods, like VoxelNeXt, FSDv2, HEDNet are missing and are not compared in Table 1.**
>
>     Thanks for the suggestions.
>     We mainly consider tracklets from GT, CenterPoint, and FSD in Table 1 because we used their tracklet proposals on the training set to train our model.  Basically, our method can generalize to other detector without retraininig.
>
>     The following table presents occupancy completion results obtained by directly applying our trained model to the tracklet proposals generated by FSDv2 on the testing data.
>     | Tracklet Inputs | IoU % | mIoU (track) % | mIoU (box) % |
>     | --- | --- | --- | --- |
>     | FSD v2 (no train) | 61.41 | 47.69 | 60.76 |
>     | FSD  | 62.84 | 54.12 | 61.58 |
>     | CenterPoint  | 57.99 | 44.94 | 55.10 |
>
>
>     Due to better detection, our method with FSDv2 still outperforms the version with CenterPoint even **without retraining**. However, it performs slightly worse compared to using FSD tracklets, despite FSDv2 having better detection results than FSD. This indicates that significant detection improvements generally lead to better shape completion (FSDv2 vs. CenterPoint). However, for detectors with similar performance (e.g., FSD vs. FSDv2), improved detections do not necessarily guarantee better shape completion without retraining. Retraining our method using training proposals generated by FSDv2 may address this issue.
>
>     We will add these results to the Table 1 and discuss the findings in our revised manuscript.
>
>
>
>     Besides, we’ve conducted a detection experiment where we applied our method to 1-frame FSDv2 without retraining.
>
>     The following table demonstrates that our method with a stronger detector continues to show detection improvement even without retraining.
>
>     | Model | Vehicle L1 mAP/mAPH | Vehicle L2 mAP/mAPH |
>     | --- | --- | --- |
>     | FSDv2 | 79.8/79.3 | 71.4/71.0 |
>     | FSDv2 +Ours | 83.2/82.7 | 75.2/74.7 |
>
>     We will include these results in our revised Table 2
>     We will also include VoxelNeXt, FSDv2, HEDNet for a more comprehensive comparison. In fact, our method based on single-frame FSD outperforms all these mentioned methods by noticeable margins.
>
>
>
>
>
> 4. **Typos/mis-leading descriptions. For example, ‘Tab. 5.4’ on line 351 -> ‘Tab. 3’.**
>
>     Thank you for the thorough review. We will correct the typos and conduct meticulous proofreading for our revised submission.

---

> > ### Comment · Reviewer_nYQd · 2024-08-13
> > **Post rebuttal**
> >
> > Thanks for the response. My concerns regarding the motivation and computation cost have been well-addressed, therefore I raise my rating to weak accept.

---

### Official Review · Reviewer_GRiU · 2024-07-12

**Soundness:** 4
**Presentation:** 4
**Contribution:** 3
**Rating:** 6
**Confidence:** 5

**Summary:**

In this work, the authors propose a novel task called object-centric occupancy.
It extends the 3D detected bounding box representation to provide a more detailed description of the internal object shape.
The method provides higher voxel resolution in large scenes by focusing on foreground objects only.
It not only achieves state-of-the-art performance on shape completion but can also help refine the object detection tasks on the Waymo Open Dataset (WOD).

**Strengths:**

- The motion of the proposed task is clear, and the task itself shows good potential in scene understanding. It can enhance 3D detection results even at a far distance.
- The extensive ablation studies validate each contribution. Various detector results with different settings help prove the robustness of the proposed methods.
Using implicit representation from a 3D reconstruction task to complete shapes is neat and interesting. It will be interesting to see how this work can be applied to popular 3D Gaussian representation.

**Weaknesses:**

- The experimental results are only obtained on the Waymo Open Dataset. It will be nicer to conduct the experiments on nuScenes or Argoverse 2 to validate its robustness for different datasets.
- Although the authors say it is a new task, so there are no learning baselines for shape completion, it will be interesting to compare the results with other scene occupancy methods. So that we can see the flaws of using coarse resolution quantitatively.

**Questions:**

- The extrapolated results of shape completion are interesting, showing that it can achieve a performance similar to that of using GT boxes. Will it also help with 3D Object Detection results?

**Limitations:**

Yes

---

> ### Author Rebuttal · Authors · 2024-08-03
>
> 1. **The experimental results are only obtained on the Waymo Open Dataset. It will be nicer to conduct the experiments on nuScenes or Argoverse 2 to validate its robustness for different datasets.**
>
>      Thanks for the suggestions. Currently, we are not able to train/test our method on nuScenes or Argoverse 2 since we only prepared object-centric occupancy labels on Waymo. We will support nuScenes or Argoverse2 after our occupancy labels on Waymo is released.
>
> 2.  **Although the authors say it is a new task, so there are no learning baselines for shape completion, it will be interesting to compare the results with other scene occupancy methods. So that we can see the flaws of using coarse resolution quantitatively.**
>
>     Thank you for the suggestion. To the best of our knowledge, no existing scene-level occupancy method has published results on Waymo Open dataset. We will release the results when their results on Waymo are published.
>
>
> 3. **The extrapolated results of shape completion are interesting, showing that it can achieve a performance similar to that of using GT boxes. Will it also help with 3D Object Detection results?**
>
>     Our detection already benefits from the extrapolated shape information, as the shape embedding used for occupancy decoding is also fused into the detection feature. Compared to explicitly using the generated occupancy, incorporating the feature directly ensures minimal information loss.

---

### Official Review · Reviewer_fcuc · 2024-07-12

**Soundness:** 3
**Presentation:** 3
**Contribution:** 3
**Rating:** 6
**Confidence:** 4

**Summary:**

The manuscript introduces the idea of representing the shape of objects at higher fidelity (and independent of) the rest of the scene. This is explored in the context of autonomous vehicles research on 3d car detection and representation. The proposed model regresses a shape code and an updated 3d bounding box from a 3D bounding box tracklet (derived from any other algorithm) and the points included in it. The shape code can be queried for occupancy to produce a full shape representation during inference. The proposed approach is able to infer complete shapes from partial inputs and the updated 3D BBs improve the input 3D BBs.

**Strengths:**

The proposed approach is relatively straightforward, effective and well motivated. This makes it reusable for other works and the paper more reproducable.

The manuscript is well written and the illustrations help convey the message and improve understanding of the written parts.

The evaluation is comprehensive and the sensitivity studies are well chosen and help motivate architecture and training choices. In particular it is great to see that the addition of the high resolution shape code and updated 3D BB does lead to substantial performance improvements on the 3D BB detection task (especially for far away OBBs). And that the shape code (if given the GT OBB) does produce a high IoU occupancy grid even if the input 3D BBs are subpar (table 1).

**Weaknesses:**

The manuscript's related work section misses out on an existing related field of 3D CAD model retrieval (which also produces complete shapes) and shape regression from RGB (and depth data) in indoor scenes. Relevant related works include:
    - Scan2CAD https://openaccess.thecvf.com/content_CVPR_2019/papers/Avetisyan_Scan2CAD_Learning_CAD_Model_Alignment_in_RGB-D_Scans_CVPR_2019_paper.pdf
    - SLAM++ https://www.doc.ic.ac.uk/~ajd/Publications/salas-moreno_etal_cvpr2013.pdf
    - FroDO https://openaccess.thecvf.com/content_CVPR_2020/papers/Runz_FroDO_From_Detections_to_3D_Objects_CVPR_2020_paper.pdf

I would have wanted to see a few renderings of the shape codes; This would support the claim that the model learns to complete shapes. The appendix has a few but the visualizations are hard to understand without a better renderer. Some kind of shading or edges for the 3D voxels are essential to see any kind of depth and thus shape (Fig 6 and 7). Extracting a mesh using marching cubes at the 0.5 isolevel might also work.

**Questions:**

The model takes in a series of 3D BBs and outputs one updated 3D BB - at which timestamp is this 3D BB output? The latest?

**Limitations:**

The limitation of rigid objects only is addressed in the appendix. This means humans are not supported for example.

---

> ### Author Rebuttal · Authors · 2024-08-03
>
> 1. **Missing related works**
>
>     Thank you for the suggestion. We will include a discussion of these works in our revised version and provide a more thorough related works section.
>
> 2. **Renderings of the shape codes;**
>
>      In the uploaded PDF, we’ve included several renderings. These renderings were obtained by applying marching cubes to the decoded volumetric grids using a level of 0.5, as suggested. The renderings demonstrate that our method can complete shapes even when the current point cloud is extremely sparse.
>
>     Due to the use of 0.2m voxel size, the resolution of our predicted occupancy may not support high-quality rendering. For example,  the resolution for a typical sedan (let's assume its dimensions are 4.5m* 1.8m * 1.4m) under our voxel size is 23 * 9 * 7. In contrast, common shape completion methods typically use a resolution of 128 x 128 x 128 or higher to facilitate high-quality rendering.
>
>     It should be noted that for our purposes, high-quality rendering is not required. Although the selected voxel size of 0.2 meters may not provide highly detailed rendering, it is sufficient for downstream driving tasks and ensures computational affordability.
>
> 3. **The model takes in a series of 3D BBs and outputs one updated 3D BB - at which timestamp is this 3D BB output? The latest?**
>
>     The behavior depends on the context. During inference, we only need to output the latest one. However, during training, we can simultaneously obtain bounding box outputs at all timestamps with just one forward pass using a causal attention mask. We compute the detection loss over all these bounding boxes to facilitate training. We will clarify this in our revised version.

---

> > ### Comment · Reviewer_fcuc · 2024-08-09
> >
> > I appreciate the authors responses. The attached shape renderings are from the same viewpoint as the input pointcloud as far as I can tell. This shows "weak completion" - interpolating between points and maybe some extrapolation. The real question is what the back side of those shapes look like "strong completion".
> >
> > I looked at the other reviews as well and did not see anything that would change my rating so far.

---

> > > ### Author Response · Authors · 2024-08-10
> > >
> > > Thank you for the quick response. As shown in Figure 10 of our appendix, our method demonstrates a strong ability for shape completion. Even with extremely sparse input points at an early timestamp (the first row in Figure 10), our method effectively completes the shape in a way that aligns with the sparse point observations. This level of completion cannot be achieved through simple interpolation or extrapolation. Our model’s success in this regard is due to its ability to learn the complete shape distribution from the training data.
> > >
> > > However, as with any learning-based approach, our method's performance can be constrained by the quality of the annotations. Since our ground truth shapes are generated by aggregating points across real object sequences, the back side of an object is often 'unobserved' (see Figure 7), meaning that most of the back-side voxels are not supervised during training.
> > >
> > > Fortunately, there is a simple yet effective way to mitigate this issue. Given the symmetry of many objects, we can fill the 'unobserved' voxels with their mirrored counterparts. Retraining our model using these mirror-aided ground truths can significantly enhance its ability to complete shapes on the back side. However, we did not employ this strategy in our main paper, as it might compromise the authenticity of the shape annotations.

---

> > > > ### Comment · Reviewer_fcuc · 2024-08-11
> > > >
> > > > Thanks for reminding me of those results in the appendix. Those are indeed strong completion results. I also see the data problem of obtaining complete shapes from partial observations.

---

> > > > > ### Author Response · Authors · 2024-08-12
> > > > >
> > > > > Thank you for recognizing the strength of our method in achieving strong shape completion. We greatly appreciate your positive feedback on this aspect of our work. Given this recognition, we kindly ask if you might reconsider your score, as it would greatly support the value of our contributions.

---

> > > > > > ### Comment · Reviewer_fcuc · 2024-08-12
> > > > > >
> > > > > > I have looked at the other reviews again and to me the main other problems raised by the other reviewers have also been addressed sufficiently. It is great to also see that the additional occ head helps detection accuracy (even if only slightly) and that the model can to some degree generalize over different detection front ends without retraining. These additional facts convince me that to raise my rating to weak accept.

---

### Official Review · Reviewer_aQAQ · 2024-07-13

**Soundness:** 2
**Presentation:** 3
**Contribution:** 2
**Rating:** 5
**Confidence:** 4

**Summary:**

This paper addresses the limitations of 3D object bounding box representations in autonomous driving by introducing object-centric occupancy. It uses an implicit shape decoder to manage dynamic-size occupancy generation. The method demonstrates robust performance under noisy conditions, significantly enhancing detection results in the Waymo Open Dataset.

**Strengths:**

1. The presentation is well-executed, with figures and charts effectively aiding reader comprehension.
2. The overall performance is impressive, demonstrating significant improvements across multiple baselines.

**Weaknesses:**

1. Creating detailed occupancy for each object seems unnecessary. In most downstream tasks in autonomous driving, using bounding boxes (bboxes) is sufficient.
2. The performance improvement primarily stems from temporal feature fusion, which lacks significant technical innovation.
3. It is unclear whether the loss on occ heads in Fig. 4 enhances detection performance. The authors should compare detection performance with and without occ heads after obtaining the Shape Emb. Z to determine if occ heads contribute to learning useful features, such as yaw estimation.

**Questions:**

See weaknesses.

**Limitations:**

The authors discuss the limitation concerning non-rigid objects, which is indeed a constraint. They could start by exploring whether reconstructing the noisy occupancy of non-rigid objects improves detection performance.

---

> ### Author Rebuttal · Authors · 2024-08-03
>
> 1. **Creating detailed occupancy for each object seems unnecessary. In most downstream tasks in autonomous driving, using bounding boxes (bboxes) is sufficient.**
>
>     We respectively disagree with this statement. As highlight in our introduction,  using bboxes alone “fails to capture the intricate details of objects’ shape, particularly for objects with irregular geometries”. As illustrated in Fig. 1, the bbox of the crane inevitably includes unoccupied space, which is however could be the drivable space for the ego-car. Relying solely on bboxes limits the downstream planner’s ability to leverage this free space effectively. Our uploaded PDF also includes samples that showcase the advantages of object-centric occupancy over bounding boxes in representing complex shape structures.
>
> 2. **The performance improvement primarily stems from temporal feature fusion, which lacks significant technical innovation.**
>
>     Indeed, our contribution does not focus on improving detection performance. As indicated in our title, we aim to take a step 'toward more flexible 3D perception,' achieved by learning an object-centric occupancy representation. This object-centric nature allows us to aggregate temporal information from a very long sequence (e.g., 200 frames) while maintaining an affordable computational cost.  Compared to previous long sequence methods, our method can additionally output object occupancy within each bbox to support precise downstream driving tasks. Additionally, with a simple temporal fusion strategy, our method surpasses previous state-of-the-art approaches in online detection, demonstrating the effectiveness of our strategy.
>
> 3. **It is unclear whether the loss on occ heads in Fig. 4 enhances detection performance. The authors should compare detection performance with and without occ heads after obtaining the Shape Emb. Z to determine if occ heads contribute to learning useful features, such as yaw estimation.**
>
>     Thank you for the suggestion. We removed the OCC head from our full model and trained the model using only the detection loss. The results are presented in the table below. A noticeable drop in detection performance is observed when the OCC decoder is removed.
>
>     | Model | L1 mAP/mAPH |  L2 mAP/mAPH |
>     | --- | --- | --- |
>     | No Occ Dec | 81.1 /80.4 | 73.0/ 72.3 |
>     | Ours | 82.8/82.3 | 74.8/74.4 |

---

> ### Comment · Reviewer_aQAQ · 2024-08-13
>
> I have read all the reviews and authors' responses. My concern has been well addressed. so I raise my rating to 5.

---

### Author Rebuttal · Authors · 2024-08-04

We would like to express our sincere gratitude to the reviewers for their thorough and thoughtful review of our paper. We are encouraged to learn that all reviewers found our paper well-written and recognized its impressive performance. We also extend our thanks to reviewers **fcuc**, **GRiU**, and **nYQd** for appreciating the novelty of our contributions.

Below, we further clarify our motivation and contributions in this work. Additionally, we provide individual responses to address the specific questions and comments raised by each reviewer.  The uploaded PDF includes several shape code renderings requested by **Reviewer fcuc**, demonstrating our model's shape completion capability and stronger representation for complex shapes.

**Motivation and Contribution**

   Our motivation is to support more flexible downstream driving tasks. And object-centric occupancy is the way we achieve this goal. Compared to traditional Bbox representation, object-centric occupancy provides a more precise representation of the obstacles and drivable space.

   To support the development of object-centric occupancy, we 1) annotated an object-centric occupancy dataset.  2) presented a robust sequence-based network for effective occupancy completion via implicit decoding. 3) showed our method also improved detection performance.
   The generated occupancy, along with the improved detection results, provides a more accurate space representation, which supports enhanced planning and control.

---

### Decision · Program_Chairs · 2024-09-25

**Decision:**

Accept (poster)

**Comment:**

Post-rebuttal, all four reviewers are in favor of acceptance. The AC has examined the paper, the reviews, rebuttal, and discussion and concurs. Since the rebuttal was critical for the paper's acceptance, the AC urges the authors to carefully incorporate all information from the rebuttal into the final version of the paper.